# Use of CPAP Ventilation in Non-ICU Wards May Influence Outcomes in Patients with Severe Respiratory COVID-19

**DOI:** 10.3390/medicina60040582

**Published:** 2024-03-31

**Authors:** Josip Brusić, Aron Grubešić, Filip Jarić, Tin Vučković, Andrica Lekić, Alan Šustić, Alen Protić

**Affiliations:** 1Department of Anesthesiology, Intensive Care and Pain Treatment, Clinical Hospital Center Rijeka, Krešimirova 42, 51 000 Rijeka, Croatia; josip.brusic@kbc-rijeka.hr (J.B.); alen.protic@medri.uniri.hr (A.P.); 2Department of Nursing, Faculty of Health Studies, University of Rijeka, Viktora Cara Emina 5, 51000 Rijeka, Croatia; 3Department of Hematology, Clinical Hospital Center Rijeka, Krešimirova 42, 51000 Rijeka, Croatia; 4Department of Internal Medicine, Faculty of Medicine, University of Rijeka, Braće Branchetta 20, 51000 Rijeka, Croatia; 5Faculty of Medicine, University of Rijeka, Braće Branchetta 20, 51000 Rijeka, Croatia; filip.jaric@gmail.com (F.J.); tinvuckovicyt@gmail.com (T.V.); 6Department of Basic Medical Sciences, Faculty of Health Studies, University of Rijeka, Viktora Cara Emina 5, 51000 Rijeka, Croatia; andrica.lekic@uniri.hr; 7Department of Anesthesiology, Reanimatology, Emergency and Intensive Medicine, Faculty of Medicine, University of Rijeka, Tome Strižića 3, 51000 Rijeka, Croatia

**Keywords:** Acute Respiratory Distress Syndrome (ARDS), Continuous Positive Air Pressure (CPAP), COVID-19

## Abstract

*Background and Objectives*: The COVID-19 disease has significantly burdened the healthcare system, including all units of severe patient treatment. Non-intensive care units were established to rationalize the capacity within the Intensive Care Unit (ICU) and to create a unit where patients with Acute Respiratory Distress Syndrome (ARDS) could be treated with non-invasive Continuous Positive Air Pressure (CPAP) outside the ICU. This unicentric retrospective study aimed to assess the efficacy of NIV Treatment in Patients of the fourth pandemic wave and how its application affects the frequency and mortality of ICU-treated patients at University Hospital Rijeka compared to earlier waves of the COVID-19 pandemic. Furthermore, the study showcases the effect of the Patient/Nurse ratio (P/N ratio) on overall mortality in the ICU. *Materials and Methods*: The study was conducted on two groups of patients with respiratory insufficiency in the second and third pandemic waves, treated in the COVID Respiratory Centre (CRC) (153 patients). We also reviewed a cohort of patients from the fourth pandemic wave who were initially hospitalized in a COVID-6 non-intensive unit from 1 October 2021 to 1 November 2022 (102 patients), and some of them escalated to CRC. *Results*: The introduction of the CPAP non-invasive ventilation method as a means of hypoxic respiratory failure treatment in non-intensive care units has decreased the strain, overall number of admissions, and CRC patient mortality. The overall fourth wave mortality was 29.4%, compared to the 58.2% overall mortality of the second and third waves. *Conclusions*: As a result, this has decreased CRC patient admissions and, by itself, overall mortality.

## 1. Introduction

The first COVID-19 case in Croatia was confirmed in Zagreb, on 25 February 2020 [1].

In further course, Croatia experienced its first wave of COVID-19 from 15 April 2020 to 18 May 2020 (5 weeks), where 95 deaths were reported out of 2224 confirmed cases (mortality 4.271%).

The second and third COVID-19 waves in Croatia started on 28 September 2020, and lasted until 23 May 2021. During the second and third waves, there were 588,861 confirmed COVID-19 cases, out of which 13,027 died. Total mortality during the second and third waves of COVID-19 in Croatia was 2.21%. The first vaccine was scarcely introduced on 23 December 2020, mostly provided to medical staff and endangered groups, with vaccinations of the wider general population occurring at the beginning of 2021 [2].

The fourth wave of COVID-19 in Croatia began in August 2021 and lasted until the end of December 2021. The total number of confirmed cases in the fourth wave is 693,102, with 12,279 confirmed deaths, for a total mortality of 1.77% in the fourth wave [3].

Vaccinations continued to the present date, with more than 5,440,000 COVID-19 vaccine doses administered. The last reported data about vaccination status in Croatia from 26 November 2023 confirms that 55% of Croatia’s population received vaccinated with a complete primary series of COVID-19 vaccines (2,246,622 in total), while 25% of the total population got at least one booster dose (996,098 in total) [2,3].

Unfortunately, by 19 September 2022, more than 1,220,000 cases of SARS-CoV-19 virus infection were confirmed in the Republic of Croatia, with 16,837 confirmed deceased from the disease or one of its complications [2]. Primorsko-Goranska County, including the Clinical Hospital Centre Rijeka (KBC Rijeka), as the county’s central healthcare facility, was among the first healthcare facilities in Croatia to open specialized COVID-19 centers concurrently treating patients with severe forms of the disease. Depending on the severity of the patient’s condition, namely the level of respiratory deterioration, we hospitalized patients in either non-ICU wards (COVID-6 unit) or in a COVID-19 specialized ICU unit named COVID Respiratory Center (CRC).

The sudden exposure of Primorsko-Goranska County and the City of Rijeka to the newly mutated virus is to be explained by the large influx of workers that have returned from Italy, especially from two regions with a rapid increase in cases of locally acquired infection: Lombardy and Veneto. The two regions have become the epicenters of Italy’s epidemic and, therefore, the primary geographic source of infection for Primorsko-Goranska County and Croatia. With the advent of the new disease, a need for creating a treatment algorithm has occurred, which will generate the most significant chance for patient survival. A statistics-based treatment algorithm was created based on the data acquired from the University Hospital Rijeka’s integrated hospital information system. In relation to the central pathophysiological mechanism of ARDS, which is the foundation of the clinical presentation in severe patients, the primary method of treatment is ventilatory support and mechanical ventilation.

The aim of the primary data collection and this study was to confirm that CPAP introduction in the University Hospital Rijeka non-ICU unit reduced patient admission to the ICU, which had an effect on overall COVID-19 patient mortality. The central hypothesis is that reduced patient admission to the ICU consequentially spares the patient-to-nurse (P/N) ratio in the ICU, which could affect the overall mortality reduction.

## 2. Materials and Methods

The aim is to compare the mortality of COVID-19-positive patients treated in earlier pandemic waves (2nd and 3rd waves) with the mortality of later pandemic waves (4th wave). In earlier pandemic waves, patients with respiratory failure were exclusively treated in the ICU (CRC), whereas in later pandemic waves (4th), some patients were able to be initially treated with CPAP non-invasive ventilation in the non-ICU (Qmed twin-dual oxygen flowmeter, Italy). The indication for CRC admission in 2nd and 3rd pandemic wave patients after the COVID unit or emergency unit assessment was SO_2_ < 90% (after the 10 L/min O_2_ administration via Venturi mask) and tachypnoea >30/min, with significant respiratory fatigue. In patients with identical clinical parameters, during the 4th pandemic wave, a CPAP mask or helmet was initially applied in non-ICU COVID-19 units without primary transfer to the CRC. Of course, if treatment with a CPAP mask or helmet failed, tracheal intubation with mechanical ventilation and transfer to the CRC occurred. The treatment protocol did not change from the 2nd to the 4th wave, and the same ICU team was with patients through that period.

Materials used to conduct this study include data on two groups of patients:A group of 153 (one hundred and fifty-three) COVID-positive patients in the 2nd and 3rd pandemic waves, from 1 September 2020 to 4 January 2021, treated in a specially formed Intensive Care Unit—CRC at the University Hospital Rijeka with invasive mechanical ventilation (IMV) and NIV application. The aforementioned group will be labelled “CRC (2nd and 3rd Wave)”. The inclusive criteria were CRC admission due to worsening of the COVID-19 clinical presentation in units where oxygen was supplied via mask with a maximal flow of 10 L/min and the above-mentioned clinical status.A group of 102 (one hundred and two) COVID-positive 4th wave patients, from 1 October 2021 to 1 January 2022, initially treated in COVID 6 Non-ICU (Infectiology/Internal Isolation Unit/Ward). Respiratory failure was treated with oxygen supplementation with up to 10 L/min flow and, with the advent of the 4th wave, CPAP non-invasive ventilation, either helmet CPAP or full-face Venturi masks (Qmed twin-dual oxygen flowmeter, Italy). The inclusive criteria were a documented application of CPAP. A portion of patients required a transfer to the CRC, with advanced ventilation therapy, either IMV or NIV, due to their worsening clinical status. The aforementioned group will be labelled as “COVID 6/CRC (4th wave)”.

In addition to the data on the type of mechanical ventilation and the treatment outcome, the following data have been gathered: age, sex, and comorbidities (arterial hypertension, heart condition, diabetes mellitus).

A mortality analysis concerning the P/N ratio during the 2nd, 3rd and 4th waves in the CRC unit has been conducted to analyze the P/N ratio on individual pandemic days.

The patients’ data from corresponding units were collected with the help of KBC´s Integrated Hospital Information System and Microsoft Excel Software (Microsoft Office—version 16.78.3) during the 2020–2022 time frames mentioned above. The study was conducted in accordance with the approval of the University Hospital Rijeka Ethical Committee. The data were analyzed using Statistica Software (Version 13.0.0.15, TIBCO Software Inc. USA) and Microsoft Excel (version 16.78). The threshold for statistical significance was *p* ≤ 0.05. Qualitative data were showcased with absolute (frequency) and relative (percentage) frequency. The difference in frequencies between groups was assessed using the chi-square test. The difference between the numeric variables (age) was tested with parametric and non-parametric tests for independent samples, depending on the data type and distribution.

## 3. Results

The first group of patients consisted of a total of 153 patients, 21 of whom were vaccinated (a total of 13.7%). The second group of patients consisted of a total of 102 patients, 24 of which were vaccinated (a total of 23.5%).

There is a statistically significant difference between the outcomes of the CRC (second and third waves) and COVID-6/CRC (fourth wave) groups (*p* < 0.001), meaning there is a significantly lesser number of deceased patients in the COVID 6/CRC (fourth wave) group (Table 1).

When considering outcomes during the fourth wave, 47% (48 total) of patients in the COVID 6/CRC group had further respiratory deterioration while on CPAP and were transferred from the COVID-6 unit to the CRC for mechanical ventilation and intensive care treatment. Despite intensive treatment, 29% (14 total) of the patients in the same group later died of CRC.

There is no statistically significant age difference between the COVID 6/CRC (fourth wave) and the CRC (second and third waves) patients (*p* = 0.776).

There is no statistically significant age difference between the deceased patients between the COVID 6/CRC (fourth wave) and the CRC (second and third waves) group (*p* = 0.067) (Table 2).

There is no statistically significant difference in sex distribution between the COVID 6/CRC (fourth wave) and the CRC (second and third waves) group, (*p* = 0.099).

There is a statistically significant difference between the deceased COVID 6/CRC (fourth wave) and the CRC (second and third waves) patients, where a significantly larger portion of deceased male patients has been observed (*p* = 0.036) (Table 3).

There is no statistically significant difference in the occurrence of diabetes mellitus, arterial hypertension and heart diseases between the deceased patients of the COVID 6/CRC (fourth wave) and CRC (second and third waves) group (Table 4).

There is a significant positive correlation between a greater P/N Ratio and mortality (*p* < 0.001, R^2^ = 0.4795) (Figure 1).

It is clearly shown how the P/N Ratio and mortality are in accordance with the pandemic waves (Figure 2).

## 4. Discussion

The surge in COVID-19 patients needing hospitalization prompted a significant response within the healthcare system. There was an increased demand for hospital beds, which raised the need for making well-thought-out decisions about how to admit patients to hospitals. In our specific case, at the University Hospital Rijeka, patients suffering from ARDS were treated during the second and third waves of the COVID-19 pandemic [4]. These patients were exclusively cared for in the CRC, and IMV was predominantly used, with occasional instances of NIV. As the number of infected individuals and patients with breathing difficulties rose, there was a decrease in the ratio of medical technicians/nurses to COVID-19 patients. This issue is noted in the literature as a contributor to the increased mortality rates from respiratory problems during the COVID-19 pandemic [4,5]. Alongside the staffing problem, challenges arose related to inadequate equipment and space to accommodate the growing demands on the CRC. Given these challenges, before the onset of the fourth wave, an initiative was taken to explore the viability of early implementation of the CPAP method for NIV in COVID-19 wards that were not intensive care units. By educating doctors and medical technicians who are not primarily ICU staff and by acquiring CPAP ventilation devices, the non-invasive CPAP method was utilized during the fourth wave of the COVID-19 pandemic. This approach was employed when a patient’s condition worsened despite receiving oxygen therapy at a rate of 10 L per minute via an oxygen mask. The aim was to use non-invasive CPAP ventilation as a means to prevent patients from needing admission to the CRC.

Before the establishment of non-intensive COVID-19 wards, patients experiencing moderate to severe acute hypoxemic respiratory failure were cared for in the CRC. This posed risks due to the treatment of respiratory insufficiency through NIV or IMV.

The initial metabolic response to the lack of oxygen in the tissues is the activation of anaerobic glycolysis and an increase in the concentration of lactate and ketones. Lipid peroxidation, especially in nerve cells, is catalyzed by iron released from hemoglobin, transferrin and ferritin, whose release is induced by tissue acidosis and free oxygen radicals. Ferroptosis-inducing factors can directly or indirectly affect glutathione peroxidase through various pathways, resulting in a decrease in the antioxidant capacity and the accumulation of lipid reactive oxygen species (ROS) in the cells, ultimately leading to oxidative cell stress and, finally, death [6].

Based on previous studies where study populations were intubated and on MV, we presumed that COVID-19 exhibits distinct characteristics compared to traditional ARDS, such as relatively intact respiratory mechanics, although having a similar severity of shunt (PaO2/FIO2) and CT abnormalities. This supports treating patients at risk of progression with CPAP early to prevent MV [7].

Furthermore, previous studies indicate that some patients with moderate-to-severe acute hemodynamic resuscitation (AHRF) caused by COVID-19 pneumonia may benefit from high-flow continuous positive airway pressure (CPAP), even when gas exchange is present, and CT findings are often associated with the need for intubation or extracorporeal oxygenation in normal acute respiratory distress syndrome (ARDS) [8].

Moreover, as patient numbers increased and medical technician/nurse availability decreased, mortality rates from acute respiratory insufficiency in intensive care units rose as well [6,7]. The purpose of this retrospective study conducted at a single centre was to assess the impact of using the CPAP mode of NIV in non-intensive COVID-19 wards. The study aimed to understand how CPAP usage influenced overall mortality and whether it had an effect on reducing admissions to the CRC.

NIV, specifically the CPAP method, holds a significant place in the treatment of ARDS [8,9,10]. Clinical studies have supported its effectiveness as a ventilation support modality for promising patients rather than a substitute for IMV [9,11,12]. These studies have shown that using CPAP mode for NIV to treat ARDS is justified compared to traditional oxygen therapy. In a randomized clinical trial involving 1273 COVID-19 patients with acute hypoxic respiratory failure, those treated with CPAP showed a notably lower rate of ICU transfer than patients receiving conventional oxygen therapy alone (55.4% vs. 62.9%). This same study demonstrated reduced endotracheal intubations and 30-day mortality among CPAP-treated patients compared to those treated solely with conventional oxygen therapy before intubation (36.3% vs. 44.4%) [13]. The early application of CPAP during the fourth wave proved effective in our patients, as it was administered following significant deterioration. In the second and third waves, this deterioration would have been an indication for admission to the CRC, an already burdened department dealing with severe forms of ARDS and a reduced number of medical staff. A statistically significant survival difference was demonstrated between patients who used CPAP (COVID 6/CRC group) and those who did not (CRC second and third waves) (Table 1). When considering outcomes during the fourth wave, 47% (48 total) of patients had further respiratory deterioration while on CPAP and were transferred from the COVID-6 unit to the CRC for mechanical ventilation and intensive care treatment. Despite intensive treatment, 29% (14 total) of the patients in the same group later died in CRC.

Ultimately, the overall mortality of patients treated with CPAP–NIV on non-intensive wards during the fourth wave, regardless of CRC admission (COVID 6/CRC group), was 29%, while the mortality rate of CRC patients from the second and third waves was 58% (Table 1). These data represent an improvement in survival and the number of hospitalized patients, both in intensive care units (CRC) and in non-intensive care units (COVID 6). The presence of comorbidities is an important factor when observing ward mortality. The most prevalent comorbidities among COVID-19 patients are arterial hypertension, diabetes, and heart disease [14,15]. Arterial hypertension is the most common among all observed patients, followed by diabetes mellitus and heart disease (Table 4), which aligns with existing literature. Arterial hypertension, as an isolated comorbidity, has shown a statistically significant impact on the survival of COVID-positive patients (Table 4). Additionally, alongside comorbidities, the age and gender of patients are important demographic data. Surviving patients are, on average, younger than deceased ones (Table 2), which is in line with similar studies [16,17]. The results show that age significantly influences the outcome of COVID-positive patients (*p* < 0.001) (Table 2), with older patients experiencing worse disease outcomes. On the other hand, we did not find a statistically significant gender effect on patient outcomes (*p* > 0.05) (Table 3). Furthermore, the study has demonstrated a significant positive correlation between the P/N ratio and mortality (*p* < 0.001) (Figure 1), confirming conclusions from several recently published studies that emphasize the lack of medical technicians as one of the most significant predictors of negative treatment outcomes for patients with severe respiratory manifestations of COVID-19 [18,19]. Additionally, our research (Figure 2) clearly shows that the shortage of medical technicians was particularly pronounced during the peak days of pandemic “waves”, which were also associated with higher mortality.

Limitations of this study are: increased vaccination rate of the fourth wave population, which had an influence on mortality; and overall improvement in managing patients with COVID-19 from the second to the fourth wave; as well as improved therapeutics and management of co-morbidities like thromboembolic prophylaxis. Nevertheless, we feel that the moderately beneficial influence of vaccination and other factors is debilitated by the increased virulence and harmfulness of the fourth wave delta variance of COVID-19, making the fourth wave significantly more challenging to overcome.

## 5. Conclusions

The utilization of CPAP ventilation mode in non-invasive COVID-19 wards can significantly reduce the burden on the highest level of intensive care medicine. This way, medical technicians, as a very important factor in intensive care medicine, do not come in an unfavorable ratio compared to the most serious COVID-19 patients and ultimately decrease mortality from severe respiratory forms of COVID-19.

## Figures and Tables

**Figure 1 medicina-60-00582-f001:**
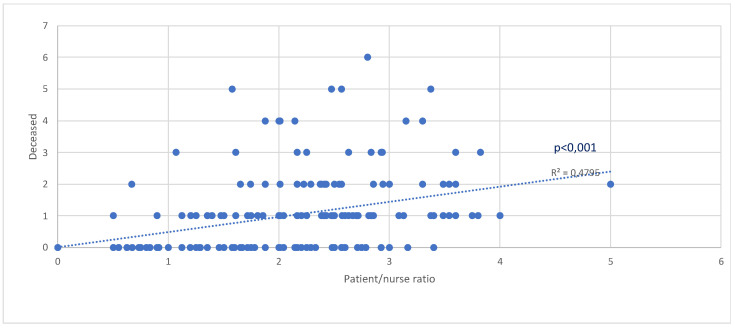
The number of deceased patients in relation to P/N Ratio.

**Figure 2 medicina-60-00582-f002:**
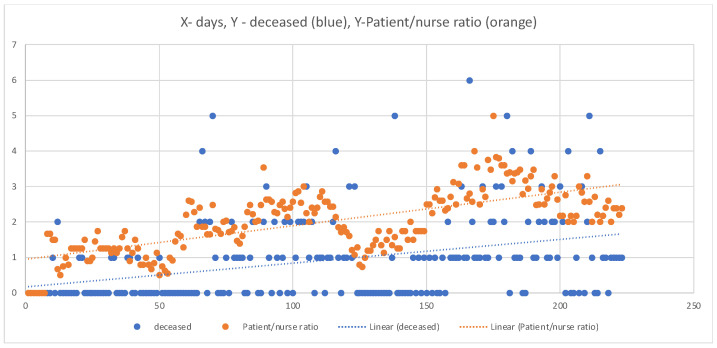
Distribution of P/N Ratio and mortality in individual Pandemic Waves. X days during the 2nd, 3rd and 4th pandemic day, Y-P/N Ratio, and the number of deceased.

**Table 1 medicina-60-00582-t001:** Overall disease outcome in surveyed patients.

Group	Deceased % (*N*)	*p*
COVID 6/CRC (4th wave) (*N* = 102)	29 (30)	<0.001
CRC (2nd and 3rd wave) (*N* = 153)	58 (89)	

**Table 2 medicina-60-00582-t002:** Age analysis of the surveyed patients.

Group	Age Mean	Standard Deviation	*p*	Age Mean Deceased	Standard Deviation Deceased	*p*
COVID 6/CRC (4th wave) (*N* = 102)	66.7	13.65	0.776	75.7	9.95	0.067
CRC (2nd and 3rd wave) (*N* = 153)	67.2	13.42		71.4	11.19	

**Table 3 medicina-60-00582-t003:** Sex distribution in surveyed patients.

Group	Female % (*N*)	Male % (*N*)	*p*	Female Deceased % (*N*)	Male Deceased % (*N*)	*p*
COVID 6/CRC (4th wave) (*N* = 102)	40 (41)	60 (61)	0.099	50 (15)	50 (15)	0.036
CRC (2nd and 3rd wave) (*N* = 153)	29 (45)	71 (108)		27 (24)	73 (65)	

**Table 4 medicina-60-00582-t004:** Comorbidity distribution (Diabetes mellitus, arterial hypertension, heart disease) in deceased patients of the two groups COVID 6/CRC (4th wave) and CRC (2nd and 3rd wave).

Group	Diabetes Mellitus Deceased % (N)	*p*	Arterial Hypertension Deceased % (*N*)	*p*	Heart Disease Deceased % (*N*)	*p*
COVID 6/CRC (4th wave) (*N* = 102)	40 (15)	0.772	60 (18)	0.133	47 (14)	0.127
CRC (2nd and 3rd wave) (*N* = 153)	35 (24)		76 (68)		29 (26)	

## Data Availability

Data are contained within the article.

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
