# Peer review of "Use of CPAP Ventilation in Non-ICU Wards May Influence Outcomes in Patients with Severe Respiratory COVID-19"

_medicina, 2024, doi:10.3390/medicina60040582_

Round 1
Reviewer 1 Report
Comments and Suggestions for Authors
I believe it is an interesting and important study, especially for countries with fewer resources
Regarding the introduction, I think it is necessary to provide information on the country's mortality separately between the 2nd and 3rd waves and the 4th wave. Was the decrease proportional to the mortality found? Was there the introduction of the vaccine? What is the country's vaccination coverage?
Objective: Clarify better the objective, whether it is to see the influence of CPAP use on overall mortality or if it is only based on the P/N rate, whether there was an increase in the rate without affecting mortality.
Methodology: Was there a change in treatment protocol between the waves? Was it the same ICU team? Which patients were vaccinated? What are the criteria for using CPAP and invasive mechanical ventilation? Did all patients who used CPAP progress to mechanical ventilation? How many only used CPAP and did not need to go to the ICU/COVID-19?
Explain better how the P/N ratio works in the ICU and non-ICU units. Were graphs 1 and 2 from which waves, were they in the ICU?
Comments on the Quality of English Language
I am not an expert in the English language, but I think the content of lines 62, 63, and 64 is somewhat unclear. The rest of text is understandable
Author Response
As attached.

Reviewer 2 Report
Comments and Suggestions for Authors
First, i would like to congratulate authors on completion of the manuscript. I do have few concers
1. With this retrospective study we cannot say for sure if cpap improved outcomes compared to IMV as the two groups are different, and there are several other factors which contribute to improvement in mortality. Example - increased vaccination, overall improvement in managing patients with covid, improved therapeutic options by the time of 4th wave, improved management of co morbidities like with strit vte prophylaxis. All these combined improve outcomes so we cannot say that cpap by itslef improved outcomes, it is mere hypothesis generating. I would recommend changing the title to more of a descriptive study looking at outcomes in croatia during the 2nd, 3rd vs 4th wave
2. please add initial demographic table, include all co morbidities, include if prior covid, include vaccination status
3. Please add limitations of your study design like mainly confounding factors as i mentioned in point 1
4. In discuss please contrast your results in more detail with other similar studies
Author Response
As attached.

Reviewer 3 Report
Comments and Suggestions for Authors
Article ``Use of CPAP Ventilation in non-ICU Wards Improves Outcomes in Patients with Severe Respiratory COVID-19''
is a retrospective study aimed to assess the efficacy of NIV treatment in patients of the pandemic COVID-19 wave and how its application affects the frequency and mortality of ICU-treated patients at University Hospital Rijeka compared to earlier waves of the COVID-19 pandemic? The introduction of the CPAP non-invasive ventilation method as a means of hypoxic respiratory failure treatment in non-intensive care units has decreased the strain, overall number of admissions, and CRC patient mortality.
It is generally known that the controlled application of oxygen therapy via NCPAP proved to be a much more effective method of treating ARDS in COVID-19 patients because patients who were treated with any mode of ventilatory support by intubation and on mechanical ventilation, had a poor prognostic course, often death and long COVID. The above can be explained by metabolic changes, ferroptosis of cells, as well as an inadequate response process on the blood vessels of the lungs, the formation of thrombosis and not primarily by an inflammatory process, in such situations the body reacts by releasing free 02 radicals.
The very idea of the work and the methodology are good, the statistics are excellent, graphically well presented. I ask the authors to make changes to the introductory part, to write about the differences between the application of NCPA and mechanical ventilation from the aspect of metabolic changes, and in the discussion, instead of repeating the results, compare their research with the research of other authors, and explain what metabolic changes occur during SARS COV- 2 infections, i.e. COVID- 19, to help my colleagues, there is an article
Jovandaric MZ, Dokic M, Babovic IR, Milicevic S, Dotlic J, Milosevic B, Culjic M, Andric L, Dimic N, Mitrovic O, Beleslin A, Nikolic J, Jestrovic Z, Babic S. The Significance of COVID-19 Diseases in Lipid Metabolism of Pregnancy Women and Newborns. Int. J. Mol. Sci. 2022; 23: 15098.
which would help explain why CPAP is better than mechanical ventilation
In addition, write the discussion with a large number of references
Author Response
As attached.

Round 2
Reviewer 1 Report
Comments and Suggestions for Authors
Dear Authors
Congratulations, I think your manuscript will be very useful for medical assistance in countries with few resources.
Thank you for accepting my suggestions.
Reviewer 3 Report
Comments and Suggestions for Authors
The article is now much better, with a clearly written introduction, a defined goal, and an explanation of why nCOPAP is more painful than mechanical ventilation.